# Specific Amino Acid Substitutions in OXA-51-Type *β*-Lactamase Enhance Catalytic Activity to a Level Comparable to Carbapenemase OXA-23 and OXA-24/40

**DOI:** 10.3390/ijms23094496

**Published:** 2022-04-19

**Authors:** Kwan-Wai Chan, Chen-Yu Liu, Ho-Yin Wong, Wai-Chi Chan, Kwok-Yin Wong, Sheng Chen

**Affiliations:** 1State Key Laboratory of Chemical Biology and Drug Discovery, The Hong Kong Polytechnic University, Hung Hom, Kowloon, Hong Kong; bill.kw.chan@connect.polyu.hk (K.-W.C.); marcus-ho-yin.wong@polyu.edu.hk (H.-Y.W.); edwardwchan@yahoo.com.hk (W.-C.C.); kwok-yin.wong@polyu.edu.hk (K.-Y.W.); 2Department of Infectious Diseases and Public Health, City University of Hong Kong, Kowloon, Hong Kong; chenyliu4-c@my.cityu.edu.hk

**Keywords:** *A. baumannii*, carbapenem resistance, carbapenem-hydrolyzing class D β-lactamases (CHDLs), OXA-51 variants, structure/function relationship

## Abstract

The chromosomal *bla*_OXA-51_-type gene encodes carbapenem-hydrolyzing class D β-lactamases (CHDLs), specific variants shown to mediate carbapenem resistance in the Gram-negative bacterial pathogen *Acinetobacter baumannii*. This study aims to characterize the effect of key amino acid substitutions in OXA-51 variants of carbapenem-hydrolyzing class D β-lactamases (CHDLs) on substrate catalysis. Mutational and structural analyses indicated that each of the L167V, W222G, or I129L substitutions contributed to an increase in catalytic activity. The I129L mutation exhibited the most substantial effect. The combination of W222G and I129L substitutions exhibited an extremely strong catalytic enhancement effect in OXA-66, resulting in higher activity than OXA-23 and OXA-24/40 against carbapenems. These findings suggested that specific arrangement of residues in these three important positions in the intrinsic OXA-51 type of enzyme can generate variants that are even more active than known CHDLs. Likewise, mutation leading to the W222M change also causes a significant increase in the catalytic activity of OXA-51. *bla*_OXA-51_ gene in A. baumannii may likely continue to evolve, generating mutant genes that encode carbapenemase with extremely strong catalytic activity.

## 1. Importance

The chromosomal *bla*_OXA-51_-like genes encoding carbapenem-hydrolyzing class D β-lactamases (CHDLs) are intrinsic to *Acinetobacter baumannii*. The role of these OXA-51-like CHDLs in mediating carbapenem resistance in *A. baumannii* is unclear. This study characterized the effect of key residues in different variants of OXA-51-like CHDLs and revealed their different contributions to catalytic activities of these enzymes, both chemically and structurally. Our finding suggests that the specific arrangement of residues in these three important positions in the intrinsic OXA-51 type of enzymes can generate variants that are even more active than known CHDLs such as OXA-23 and OXA-24/40. Our study suggests that some of the intrinsic variants of OXA-51-like CHDLs could mediate carbapenem resistance in *A. baumannii.*

## 2. Introduction

*Acinetobacter baumannii* is an important Gram-negative pathogen that often causes severe infections in immunocompromised patients in intensive care units (ICUs) [1]. According to the data from United States Center for Disease Control and Prevention, it was estimated that about 8500 *A. baumannii* infections occurred in the year 2017, of which 75% were extended-spectrum β-lactam-resistant and resulted in approximately 700 deaths [2]. Production of carbapenem-hydrolyzing class D β-lactamases (CHDLs) is one major mechanism that mediates the expression of phenotypic carbapenem resistance in clinical *Acinetobacter baumannii* strains, rendering carbapenems, the last-resort antibiotics, ineffective in treating infections caused by this important bacterial pathogen [3,4,5,6]. OXA-23, OXA-24/40, and OXA-58, the prevalent enzymes encoded by resistance genes transmissible among clinical *A. baumannii* strains, are known to be key carbapenem resistance determinants in this pathogen [3,6,7].

A chromosomally encoded CHDL enzyme produced by *A. baumannii*, namely, OXA-51, has also been recognized as a determinant of carbapenem resistance [8,9]. A considerable number of studies have been conducted on *bla*_OXA-51_-like β-lactamases, which are intrinsically encoded in *A. baumannii* chromosome and can be readily overexpressed by promoter activation through insertion sequences such as IS*Aba1* [10]. This enzyme is normally not over-expressed under its natural promoter. The low expression level of this enzyme and its weak activity have limited its ability to confer phenotypic resistance to carbapenem in *A. baumannii*. However, recent studies have shown that a dramatically increased expression of this gene driven by the IS*Aba1* insertion element can mediate the expression of significant carbapenem resistance [10]. The inability of this enzyme to cause high-level resistance despite a sharply elevated expression level is probably due to its low catalytic activity compared to OXA-23 and OXA-24/40 [10]. Many variants of OXA-51 that harbor one or more amino acid substitutions have been observed clinically; residues W222, L167, I129L, and P130 are the most common positions in which amino acid substitutions occur [11,12,13]. For example, enzymes harboring the W222G and W222L changes were named OXA-79 and OXA-200, respectively; in particular, W222L substitution has been shown to enhance the affinity for carbapenems [13]. The substitution L167V has also been commonly observed in many different subtypes of OXA-51-like enzymes, including OXA-51 (yielding OXA-219 [14]), OXA-66 (yielding OXA-82 [15]), OXA-69 (yielding OXA-107 [15]), and OXA-71 (yielding OXA-113 [16]) [11]. Despite multiple substitutions in OXA-51 variants being identified, there is a lack of knowledge on the degree of contribution of individual amino acid change to variation in catalytic the activity of OXA-51. There is also a lack of understanding of the comparative catalytic activity of different variants of OXA-51 and the major carbapenemase OXA-23 and OXA-24/40 in *A. baumannii*. In this study, we analyzed the effect of amino acid substitutions in several common enzyme variants identified in our previous study [17] to provide insight into the functional capacity of various OXA-51 variants in mediating the expression of phenotypic carbapenem resistance in *A. baumannii*.

## 3. Materials and Methods

### 3.1. Antibiotics and Media

Biapenem, imipenem, meropenem, and ertapenem were purchased from Melonepharma Co. (Dalian, China). Isopropyl β-D-1-thiogalactopyranoside (IPTG) was purchased from Santa Cruz Biotechnology Inc. (Dallas, TX, USA). Kanamycin and Luria broth (LB) were purchased from Thermofisher Scientific Inc. (Waltham, MA, USA). Muller–Hinton broth (MHB) was purchased from Becton, Dickinson and Company (Franklin Lakes, NJ, USA).

### 3.2. Genetic Analysis of OXA-51 Variants

*Acinetobacter baumannii* clinical strains were isolated from patients hospitalized in two local hospitals in Hong Kong and Henan Province, China, from 2000 to 2013. The 14 strains that only carried *bla*_OXA_ alleles but did not express other β-lactamases were chosen for analysis of their *bla*_OXA-51_ alleles in this study (Appendix A). Detailed information, including their genome sequences, was provided in our previous study [17]. These *bla*_OXA-51_ alleles were shown to mediate carbapenem resistance in these *A. baumannii* strains, in which they were over-expressed as a result of the insertion of the IS*Aba1* element in the promoter region of the *bla*_OXA–51_ operon. The amino acid sequence alignment of these OXA-51 variants is shown in Figure 1.

### 3.3. Cloning of bla_OXA_ and Site Directed Mutagenesis of bla_OXA-51_-Like Variants

The complete protein-encoding sequence of different *bla*_OXA_ genes was obtained from the clinically isolated *A. baumannii* strains described above. The signaling peptide of each protein was identified by Signal P5.0 Server. Target *bla*_OXA-51_ variant genes were amplified by PCR and cloned into a pET-15b expression vector. The primers used in PCR are listed in Table 1. The constructs were transformed into *Escherichia coli* DH5α and subjected to confirmation of the proper constructs by Sanger sequencing. The confirmed constructs of *bla*_OXA-51_-like variants were further subjected to site-directed mutagenesis. Briefly, the purified plasmid DNA was mutated using GeneArt^TM^ Site Direct Mutagenesis System from Invitrogen (Waltham, MA, USA). The procedures followed the instructions provided by the manufacturer. The primers used for constructing different mutants are listed in Table 1. The mutations generated were confirmed by Sanger sequencing. The confirmed constructs of *bla*_OXA-23_, *bla*_OXA-72_, *bla*_OXA-51_ variants and *bla*_OXA-51_-derived mutants were further transformed into *Escherichia coli* BL-21 for protein purification.

### 3.4. OXA Type β-Lactamase Purification and Enzyme Kinetic Assay

Overnight culture of *E. coli* BL-21 carrying different constructs of *bla*_OXA_ was sub-cultured, with 1% of culture inoculated into 500 mL of fresh LB broth supplemented with ampicillin and incubated at 37 °C with shaking at 250 rpm until OD600 reached 0.6. The culture was then induced by IPTG with a final concentration at 0.5 mM and allowed to further incubate at 16 °C overnight with gentle shaking. The cells were collected and lysed as described previously [18]. Briefly, the soluble fractions were passed through a Ni-NTA column. The eluted fractions containing the target proteins were pooled and concentrated with an Amicon Ultra-15 (NMWL = 10,000) centrifugal filter device. The His_6_ tag of OXA proteins was further removed and purified following the procedures described previously [19]. The purified protein was analyzed by SDS-PAGE. The purified OXA-51 variants and OXA-72 proteins were stored in PBS buffer at pH 7.0, whereas the OXA-23 protein fraction was stored in PBS buffer with 40% glycerol at pH 8.0 to prevent precipitation since its isoelectric point is close to pH 7.0. All protein fractions were aliquoted and flash-frozen by liquid nitrogen, followed by storage at −80 °C. The concentration of enzyme was measured by NanoDrop and diluted to different ratios before the experiment. Enzyme kinetic assay was conducted by measuring the hydrolytic activity of each Bla_OXA_ enzyme on ampicillin, cefotaxime, imipenem, and meropenem. The assay buffer used was 50 mM phosphate buffer supplemented with 50 mM NaHCO_3_ with pH 7.0. The reaction buffer was first mixed with meropenem of known concentration. Measurement was started immediately after the enzyme was added. The hydrolysis of antibiotics was monitored by measuring absorbance of a specific wavelength continuously for 180 s.

### 3.5. Carbapenem Susceptibility Test of Constructs Carrying bla_OXA_ Variants and Their Corresponding Mutants

Over-expression of OXA-51 by pET-15 vector in *E. coli* is not a sound system to determine the MIC of these enzymes since OXA types of carbapenemases do not confer high MIC in *E. coli*. To determine the MIC of these OXA-51 variants, we have developed a system in *A. baumannii* using the vector pBAV1K-T5-MCS-KanR, which was modified from pBAV1K-T5-GFP. The original coding sequence of GFP of pBAV1K-T5-GFP was replaced by multiple cloning sites (MCS) to generate pBAV1K-T5-MCS-KanR. Briefly, the complete protein-encoding sequence of different *bla*_OXA_ genes and their corresponding natural promoters (IS*Aba1* sequence was added to the upstream of the coding region for the constructs *bla*_OXA-51_-like and *bla*_OXA-23_, and 346bp upstream sequence was cloned with the coding region of *bla*_OXA-72_) were amplified by PCR and inserted into the pBAV1K-T5-MCS-KanR vector. Site direct mutagenesis was performed on the constructed plasmids. The primers used in PCR and mutagenesis are listed in Table 1. The constructs were all transformed into the host strains *A. baumannii* ATCC17978 and subjected to Sanger sequencing to ensure fidelity of all sequences. The susceptibility of six antibiotics, biapenem, meropenem, imipenem, ertapenem, cefotaxime, and ampicillin, to various Bla_OXA-51_-like variants, and their corresponding mutants, were tested according to CLSI guidelines [20]. Briefly, bacteria strains grown on MH agar plates were suspended in sterilized 0.85% sodium chloride solution. The turbidity of bacterial cells suspension was adjusted according to McFarland Standard No. 0.5, followed by inoculation into MH broth supplemented with serially diluted concentrations of carbapenems. All testing mixtures were then incubated at 37 °C for 16 h. The reference strains *A. baumannii* ATCC17978 and *E. coli* ATCC25922 were used as control. The susceptibility test was repeated three times to ensure consistency.

### 3.6. Structure Modelling and Analysis

The Michaelis complex structure of OXA-51 with meropenem was generated by docking the meropenem (PubChem CID: 441130) molecule into OXA-51 (PDB ID: 5L2F [11]). The structure of mutated OXA-51 containing the amino acid substitutions I129L, L167V, and W222G was generated using the Mutagenesis Wizard function of PyMOL (Schrödinger, Inc., NY, USA). Both structures were docked with meropenem using the Autodock Vina algorithm (The Scripps Research Institute, CA, USA). The output conformations were subjected to PyMOL for further analysis. The distances between different molecules were determined by the Measurement Wizard function of PyMOL.

## 4. Results and Discussion

### 4.1. Genetic Analysis of Different OXA Variants

Our previous study on the genome analysis of clinical A. baumannii strains identified several OXA-51 variants, including OXA-66, OXA-79, OXA-82, OXA-83, and OXA-99 [17]. Some of these intrinsic OXA-51 variants were over-expressed and found to confer carbapenem resistance in these clinical *A. baumannii* strains, suggesting that OXA-51 variants might exhibit comparable catalytic activity as the acquired OXA-23 and OXA-24/40 types of CHDLs, which are the major enzymes that cause carbapenem resistance in *A. baumannii* [17].

In the current study, enzyme kinetics and comparative mutagenesis experiments were performed to investigate how different mutations affect the catalytic activity of OXA-51-like CHDLs, with the OXA-23 and OXA-24/40 types CHDLs being the control enzymes. Comparative sequence analysis of these OXA-51 variants is shown in Figure 1 [21]. According to the amino acid sequence alignment, OXA-99 has the highest degree of similarity with OXA-51 among the selected enzymes, containing only two amino acid substitutions, namely, lysine to methionine at position 209, and glutamine to arginine at position 57. In OXA-66, there were six amino acid substitutions (T5A, E36V, V48A, Q107K, P194Q, D225N) compared to OXA-51. Interestingly, there were single amino acid substitutions in OXA-79, OXA-82, and OXA-83 compared with OXA-66. The alignment results suggest that OXA-79, OXA-82, and OXA-83 are variants of OXA-66 and that OXA-66 and OXA-99 are variants of OXA-51 generated through different branches of the evolutionary pathway.

### 4.2. Enzymatic Activity of Different OXA Variants

The catalytic characteristics of OXA-23, OXA-72, OXA-51, OXA-66, OXA-79, OXA-82, OXA-83, and OXA-99 against ampicillin, imipenem, and cefotaxime are shown in Table 2. All the OXA enzymes tested in this study did not exhibit any catalytic activity against cefotaxime. The result showed that the catalytic ability of OXA-66 against ampicillin was reduced by half when compared with OXA-51. However, OXA-99 exhibited about 1.7-fold increase in activity compared to OXA-51 (Table 2). Among the OXA-66 variants, the OXA-79 exhibited the highest activity against ampicillin, which represented a 62.56-fold increase compared to OXA-51, and threefold higher than that of OXA-72 (Table 2). The reduced *K_m_* to ampicillin of OXA-79 indicated that substitution of tryptophan by glycine at position 222 resulted in higher affinity to ampicillin. This finding suggests that the bulky W222 side chain could hinder the ampicillin molecule from approaching the S80 active site. Replacement of this residue (position 222) by a smaller glycine molecule, as in the case of OXA-79, resulted in the exposure of the S80 active site to ampicillin, which in turn resulted in an increase in *k_cat_* and reduced *k_m_*. Other variants, including OXA-82 and OXA-83, also exhibited increased activity against ampicillin when compared to OXA-66 and OXA-51. On the other hand, a comparison of the catalytic activity of these OXA enzymes on imipenem showed that OXA-72 conferred nearly twofold higher activity against imipenem than OXA-23 (Table 2).

Among the OXA-51 variants, the difference between OXA-51 and OXA-99 was similar to that of ampicillin. Among OXA-66 and its variants OXA-79, OXA-82, and OXA-83, OXA-79 was found to exhibit about 10-fold higher activity than OXA-66 in a manner similar to the difference between the activities of these two enzymes on the ampicillin substrate (Table 2). Surprisingly, OXA-82 and OXA-83 exhibited even higher activity against imipenem than OXA-79, with 21.94- and 27.25-fold increases in activity compared to OXA-51, respectively (Table 2). These findings suggest that residues I129 and L167 are important for the catalytic activity against carbapenem among the OXA-51 type enzymes. To confirm the enzyme kinetic assay results, strains carrying different *bla_OXA_* constructs were subjected to MIC tests. OXA-23 and OXA-79 were found to confer an ampicillin MIC of >2048 μg/mL in *A. baumannii* ATCC17978, which was consistent with the high enzymatic activity of OXA-23 and OXA-79 against ampicillin recorded in the enzyme kinetic assay (Table 2). The other OXA-51 variants constructs were found to confer an ampicillin MIC of 32μg/mL in ATCC17978, representing a fourfold increase. The constructs of OXA-82 and OXA-83 confer the highest imipenem MIC among the OXA-51 variants, but the value was two- and fourfold lower than that of OXA-23 and OXA-72 (Table 2). These data are also consistent with the results of the enzyme kinetic assay. All the test constructs did not confer any resistance against cefotaxime in ATCC17978, confirming that cefotaxime is not a substrate of these OXA enzymes.

An enzyme kinetic assay was performed with meropenem as substrate to further investigate the enzymatic activity of OXA-51 variants against carbapenem. Enzyme activity analysis showed that OXA-99 exhibited slightly lower catalytic activity against meropenem when compared to OXA-66 (0.86-fold). Other OXA-51 variants, including OXA-79, OXA-82, and OXA-83, were found to exhibit 3.07-, 1.33-, and 4.5-fold increases in activity when compared to OXA-66 (Table 3). To further confirm the difference between the degree of carbapenem resistance conferred by these OXA-51 variants, a MIC test of *A. baumannii* ATCC17978 carrying these variants was performed. Our previous study confirmed that ISAba1 is essential for the over-expression of blaOXA-51 and blaOXA-23 and, hence, the expression of phenotypic resistance in *A. baumannii* [17]. In this work, the corresponding constructs were linked to an IS*Aba1* promoter to facilitate the over-expression of carbapenemases. As expected, the MIC of meropenem of *A. baumannii* ATCC17978 carrying these variants is consistent with the corresponding catalytic activities encoded by these variants (Table 3).

### 4.3. Enzymatic Activity of Different OXA Mutants

To further test and confirm the degree of contribution of various amino acid substitutions to the catalytic activity of OXA-51 variants, a single-point mutation was sequentially introduced into different OXA-51 variants. The K209M substitution in OXA-66 resulted in a decreased catalytic activity compared to OXA-66, with 7.5-fold reduction in *k_cat_* and a 3.05-fold decrease in *K*_m_; these parameters represent a twofold reduction in catalytic activity, which was even lower than that of OXA-99. Consistently, MICs of various carbapenems in organisms expressing OXA-66 (K209M) were reduced to a level similar to those producing OXA-99 (Table 3). Introducing this amino acid substitution into OXA-82 to create OXA-82 (K209M) resulted in a roughly twofold reduction in *k_cat_* but had no effect on *K*_m_; nevertheless, an overall roughly twofold reduction in catalytic activity was recorded when compared to OXA-82. Consistently, the MICs of different carbapenems also decreased slightly (Table 3). These data suggested that the K209M change exhibited a variable effect on substrate catalysis in different OXA-51 variants. A similar reduction was seen in both enzyme catalysis and MICs of various carbapenem for strains producing the K209M-bearing OXA-79 enzyme (Table 3).

A comparison of the sequence of OXA-82 to OXA-66 showed that an L167V substitution in OXA-66 was responsible for converting it to OXA-82 and causing threefold and a fourfold reduction in *k_cat_* and *K*_m_, respectively, and an overall 1.33-fold increase in catalytic activity, suggesting that the L167V substitution affected the catalytic activity of OXA-66 to a certain extent. Further introduction of an additional amino acid substitution, W222G, to form OXA-82 (W222G), resulted in a slight increase in *k_cat_*, a slight decrease in *K*_m_, and an overall 1.5-fold increase in catalytic activity. The introduction of I129L to produce OXA-82 (I129L) did not affect *k_cat_* but caused a fourfold decrease in *K*_m_ when compared to OXA-82; these changes enabled the overall catalytic activity to increase up to roughly fourfold. Simultaneous incorporation of these two substitutions to form OXA-82 (W222G, I129L) led to an increase of *k_cat_* by 2.5-fold and reduction of *K*_m_ by roughly twofold; in this case, the overall catalytic activity was found to increase by roughly fivefold. When compared to OXA-99, a 7.68-fold increase in catalytic activity, a level higher than that of OXA-23 and approaching that of OXA-72, was recorded. Consistently, the MICs of different carbapenems for these mutants were found to increase accordingly (Table 3). These data suggested that substitutions L167V, W222G, and I129L could all cause an increase in the catalytic activity of OXA-66, with the effect of the substitution I129L being the most dramatic. The elevated catalytic activity of OXA-51 due to these three mutations was reported in some previous studies [12,22].

The fact that simultaneous introduction of three amino acid substitutions in OXA-66 could increase the catalytic activity to the level of common carbapenemase OXA-23 and OXA-24/40 types suggests that a number of mutations in CHDL-encoding genes could confer resistance to carbapenem in *A. baumannii*. Consistently, OXA-83, which is equivalent to OXA-66 containing the I129L change, exhibited 1.5-fold decrease in *k_cat_* and an eightfold reduction in *K*_m_ when compared to OXA-66, further confirming the role of I129L in substrate binding. Further introduction of the W222G substitution, which resulted in the formation of OXA-83 (W222G), caused a slight increase in *k_cat_* and a further slight decrease in *K*_m_, as well as a ≈1.5-fold increase in the overall catalytic activity when compared to OXA-83. Compared to OXA-99, its catalytic activity was found to have increased 8.38-fold, which is even higher than that of OXA-23 or OXA-72. These data also suggested that incorporating the W222G and I129L changes into OXA-66 resulted in slightly higher catalytic activity than that conferred by the three amino acid substitutions of L167V, W222G, and I129L. Lastly, the role of the I129L change in other variants such as OXA-99 was tested and shown to cause a roughly twofold reduction in *K*_m_ but no effect on *k_cat_*, thereby further confirming the role of I129L in enhancing substrate affinity in different variants of OXA-51 (Table 3). The MICs for these mutants of different carbapenems were consistent with the enzyme kinetics data (Table 3).

### 4.4. In Silico Modelling of Different OXA Mutants

To explain the role of amino acid substitutions in these important residues in enhancing the catalytic activity of OXA-51 variants, the OXA-51 complex with meropenem was modelled (Figure 2). The residues I129L, L167, and W222 were all located around the active site of OXA-51. Their distances with meropenem were 3.4, 6.4, and 2.9 Å, respectively, suggesting that these three residues might change the conformation of the OXA-51 active site pocket to impose steric hindrance that prevents the approach of substrates to OXA-51, rendering it less active in substrate hydrolysis (Figure 2A). The L167V change caused the distance between V167 and meropenem to decrease from 6.4 to 4.0 Å; this may explain why this amino acid substitution could result in a relief of the steric hindrance effect and an increase in substrate binding affinity, but this amino acid change may also affect substrate catalysis due to its influence in causing less optimal substrate alignment.

The short distance between W222 and meropenem may suggest a huge hindrance effect of this residue on substrate binding (Figure 2A). The W222G change is associated with extending the distance from 2.9 to 5.1 Å. Consistently, a minor effect on the catalytic activity of OXA-66 against meropenem was observed, presumably due to the combined effect of reducing substrate hindrance but at the same time affecting optimal substrate alignment by minimizing interaction with this site due to the extension of the distance between this residue and meropenem (Figure 2C). An OXA-66 (W222M) variant was created and exhibited slightly higher catalytic activity than OXA-66 (W222G). The combination of W222M with other amino acid substitutions, such as OXA-66 (L167V, W222M) and OXA-66 (I129L, W222M), led to an increase in catalytic activity by roughly twofold and eightfold when compared to OXA-66 (L167V, W222G) and OXA-66 (I129L, W222G), respectively (Table 3). The W222M change could reduce the hindrance effect and maintain hydrophobicity at this position, therefore increasing the catalytic activity of OXA-66. The result aligned with the data of Smith et al., which showed that the W222 residue occluded the active site of OXA-51 and interfered with the carbapenemase activity of the enzyme [23]. During the evolution process, the substitution of the W222 residue by other structurally simpler amino acids such as methionine or glycine could enhance the survival fitness of *A. baumannii* in the nosocomial environment. The OXA-51 W222M change can also be seen in OXA-23 and OXA-72 (Figure 1), which corresponds to the residue in OXA-23 M240 and OXA-72 M223. The affinity of meropenem to OXA-23 and OXA-72 is also much higher than OXA-51, which is reflected by their *K_m_* value (Table 3). Both enzymes are prevalent among CHDLs producing *A. baumannii*, and genes encoding such enzymes can be disseminated through plasmid exchange. On the basis of the fact that OXA-23 and OXA-72 exhibit much stronger carbapenemase activity than OXA-51, there is strong evidence that the hindrance of W222 residue plays an important role in carbapenemase activity.

The mutation that resulted in the I129L change caused the distance between residue 129 and meropenem to decrease from 3.4 to 2.8 Å, resulting in shortened interaction distance between the S80 residue and the meropenem molecules from 2.7 to 2.1 Å. The relief could therefore explain its effect on enhancing substrate binding in substrate hindrance effect and optimization of substrate interaction. This theory is consistent with the finding of a previous study in that the I129L change could transform the active site of OXA-51 into a conformation that significantly enhances its interaction with carbapenems (Figure 2C) [11]. The strong effect of the I129L substitution has also been reported in several other studies [11,12,13]. The fact that the fivefold increase of catalytic activity reported in this study was different from that of >80-fold in other studies is probably due to the difference in kinetic constants recorded for different carbapenem antibiotics [11,12]. To further prove this hypothesis, various substitutions have been created, including OXA-66 (I129A), OXA-66 (I129D), OXA-66 (I129F), OXA-66 (I129M), and OXA-66 (I129V). We showed that substitution to Val and Ala resulted in a slight reduction in MIC of meropenem. In contrast, substitutions to Asp, Phe, and Met led to a further reduction in the MIC of meropenem, with the most dramatic effect recorded in Asp substitution (Table 3). These data further confirm that a hydrophobic residue with the proper size of the side chain is important for the optimal functioning of this residue.

Most interestingly, we showed in this study that the effects of the I129L and W222G changes in OXA-66 were additive, resulting in an increase of the catalytic activity to a level higher than that of OXA-23 or OXA-72. Further addition of L167V into OXA-66 (I129L, W222G), however, did not exhibit an additive effect but caused a slight reduction in catalytic activity, suggesting that optimal arrangement of residues in these three important positions should generate a variant that is even more active than OXA-51. The introduction of W222M into this combination, OXA-66 (I129L, W222M), further reduced the *K*_m_ by roughly fourfold and caused a net increase in catalytic activity by ≈14.3-fold. Consistently, the MICs for different carbapenems for strains producing OXA-66 (I129L, W222M) ranged from 64 to >512µg/mL, which are higher than those of OXA-23 and OXA-72 (Table 3). Lastly, we also looked into the negative effect of the K209M change on the catalytic activity of OXA-51 variants. Structural analysis showed that K209 was located on one side of the four β-sheets and formed a salt bridge with the E269 residue located at the C-terminal helix. This interaction might help stabilize the β-sheets and C-terminal helix that formed the active site cavity. Consistently, amino acid substitutions at K209 were found to affect substrate catalysis instead of substrate binding, thus further supporting this hypothesis (Figure 2D).

## 5. Conclusions

In conclusion, this study tested the effect of amino acid substitutions in several variants of OXA-51 that exhibited various levels of enhancement in the catalytic activity of the enzyme. These substitutions were shown to result in structural changes around the active site of OXA-51 and therefore variation in the degree of interaction between the enzyme and its substrates. Mutational and structural analysis showed that the substitutions L167V, W222G, or I129L contributed to an increase in the catalytic activity of OXA-51 variants, with the I129L change exhibiting the highest enhancement effect. Interestingly, the W222G and I129L changes exhibited a stronger enhancement effect than that conferred by a combination of the above three amino acid substitutions, depicting an optimal arrangement of residues in these three important positions and that such arrangement could generate even a more active variant of OXA-51. Further studies on the most optimal amino acid combinations in these three positions that would produce the most active OXA-51 variants should be conducted to investigate further the structure and activity relationship of this important intrinsic carbapenemase.

## Figures and Tables

**Figure 1 ijms-23-04496-f001:**
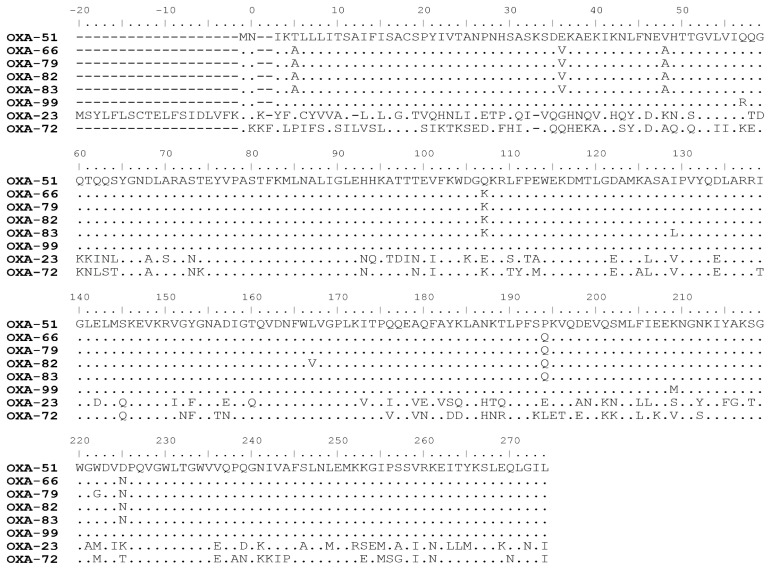
**Amino sequence alignment of OXA identified in this study.** Alignment was performed by Cluster W. Sequence of OXA-51 is included as reference sequence. OXA-23 and OXA-72 (a variant of OXA-24) were also included in the alignment to show the difference between OXA-23, OXA-72, and other OXA-51 variants.

**Figure 2 ijms-23-04496-f002:**
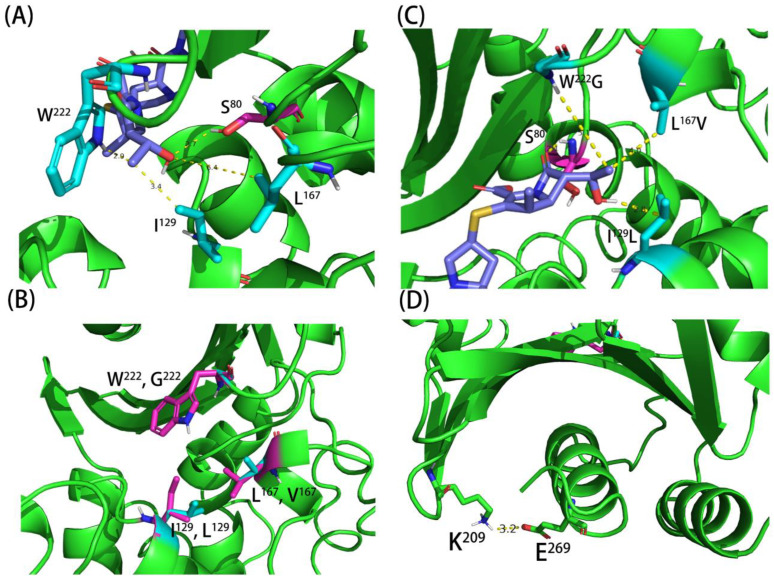
**Conformation of OXA-51 and distance between important active site residues in OXA-51 variants and meropenem.** (**A**) Distance between residue I129L/L167/W222 and meropenem. (**B**) Comparison of the effect of I129, L167, W222, L129, V167, and G222 on the confirmation of OXA-51; the residue of wild-type OXA-51 are highlighted in purple, and amino acid substitutions are highlighted in tiffany blue. (**C**) Distance between residue L129/V167/G222 and meropenem. (**D**) Salt-bridge formed by K209 and E269.

**Table 1 ijms-23-04496-t001:** Primers used in this study.

Primer	Sequence 5′ to 3	Applications
OXA23-BamHI	CGATGGATCCATGAGTTATCTATTTTTGTCGTGTACAGAG	Cloning of *bla*_OXA-23_
OXA23-NdeI	CGATCATATGTTAAATAATATTCAGCTGTTTTAATGATTTCATCAA	
OXA51-BamHI	CGATCGATCGGATCCAATCCAAATCACAGCGCTTCA	Cloning of *bla*_OXA-51_ and its variants
OXA51-NdeI	CGATCATATGCTATAAAATACCTAATTGTTCTAAGCTTTTATAAGT	
OXA72-BamHI	CGATGGATCCTCTATTAAAACTAAATCTGAAGATAATTTTCATATT	Cloning of *bla*_OXA-72_
OXA72-NdeI	CGATCATATGTTAAATGATTCCAAGATTTTCTAGCG	
IS*Aba1*-BamHI	CGATGGATCCCTAAATGATTGGTGACAATGAAGTTTTTTT	Cloning of *bla*_OXA-23_, *bla*_OXA-51_, and their variants (for Carbapenem susceptibility test)
*bla*_OXA51_-XhoI	CGATCTCGAGCTATAAAATACCTAATTGTTCTAAGCTTTTA	
*bla*_OXA23_-XhoI	CGATCTCGAGTTAAATAATATTCAGCTGTTTTAATGATTTCATCA	
*bla*_OXA72_-BamHI	CGATGGATCCCGATTCTTAGCCTCATCCCA	Cloning of bla_OXA-72_ (for Carbapenem susceptibility test)
*bla*_OXA72_-XhoI	CGATCTCGAGTTAAATGATTCCAAGATTTTCTAGCGACT	
Q57R-F	GTGTTTTAGTTATCCGACAAGGCCAAACTCA	Mutagenesis of Q57R
Q57R-R	TGAGTTTGGCCTTGTCGGATAACTAAAACAC	
I129L-F	ATGAAAGCTTCCGCTCTTCCAGTTTATCAAG	Mutagenesis of I129L
I129L-R	CTTGATAAACTGGAAGAGCGGAAGCTTTCAT	
L167V-F	GTCGATAATTTTTGGGTGGTGGGTCCTTTAA	Mutagenesis of L167V
L167V-R	TTAAAGGACCCACCACCCAAAAATTATCGAC	
K209M-F	TATTCATAGAAGAAATGAATGGAAACAAAAT	Mutagenesis of K209M
K209M-R	ATTTTGTTTCCATTCATTTCTTCTATGAATA	
W222G-F	AAAAGTGGTTGGGGAGGGGATGTAAACCCAC	Mutagenesis of W222G
W222G-R	GTGGGTTTACATCCCCTCCCCAACCACTTTT	

**Table 2 ijms-23-04496-t002:** Enzyme kinetic constants of different OXA lactamases and MIC of ampicillin, imipenem, and cefotaxime in *A. baumannii* strain ATCC17978 carrying these OXA.

OXA-23/OXA-24/OXA-51 Variants	Kinetics Constants													
	Ampicillin					Imipenem					Cefotaxime			
	MIC (mg/L)	*k_cat_* (s^−1^)	*k_m_* (μM)	*k_cat_*/*k_m_* (μM^−1^/s^−1^)	Fold Increase	MIC (mg/L)	*k_cat_* (s^−1^)	*k_m_* (μM)	*k_cat_*/*k_m_* (μM^−1^/s^−1^)	Fold Increase	MIC (mg/L)	*k_cat_* (s^−1^)	*K_m_* (μM)	*k_cat_*/*k_m_* (μM^−1^/s^−1^)
OXA-23	>2048	6.00 ±0.34	53.1	1.129 × 10^−1^	80.91	32	0.192 ± 0.006	0.88	2.174 × 10^−1^	39.06	8	* n.d	n.d	n.d
OXA-72	1024	5.63 ± 0.53	198.4	2.839 × 10^−2^	20.34	64	0.145 ± 0.005	0.37	3.965 × 10^−1^	71.24	8	n.d	n.d	n.d
OXA-51	32	1.32 ± 0.25	945.6	1.396 × 10^−3^	1.00	1	0.268 ± 0.014	48.2	5.566 × 10^−3^	1.00	8	n.d	n.d	n.d
OXA-66	32	1.02 ± 0.37	1576	5.683 × 10^−4^	0.41	1	0.113 ± 0.004	34.3	3.291 × 10^−3^	0.59	8	n.d	n.d	n.d
OXA-79	>2048	16.61 ± 2.14	190.2	8.733 × 10^−2^	62.56	4	0.158 ± 0.003	2.9	5.533 × 10^−2^	9.94	8	n.d	n.d	n.d
OXA-82	32	3.01 ± 0.53	619.9	4.854 × 10^−3^	3.48	16	0.226 ± 0.006	1.8	1.221 × 10^−1^	21.94	8	n.d	n.d	n.d
OXA-83	32	0.87 ± 0.05	223.9	3.872 × 10^−3^	2.77	16	0.095 ± 0.001	0.63	1.517 × 10^−1^	27.25	8	n.d	n.d	n.d
OXA-99	32	5.72 ± 2.27	2370	2.413 × 10^−3^	1.73	1	0.337 ± 0.020	33.8	9.973 × 10^−3^	1.79	8	n.d	n.d	n.d
ATCC17978	8	-	-	-	-	0.125	-	-	-	-	8	-	-	-
ATCC25922	4	-	-	-	-	≤0.06	-	-	-	-	≤0.06	-	-	-

* n.d, the enzymatic activity is undetectable. -, no data. The control strains for MIC did not contain any enzyme kinetic data.

**Table 3 ijms-23-04496-t003:** Effect of specific amino acid substitutions in OXA-51 variants on their kinetic behavior against different carbapenem and carbapenem MIC in *A. baumannii* ATCC17978.

OXA-23/OXA-24/OXA-51 Variants	Amino Acid Substitutions in OXA-51/OXA-66	Kinetic Constants				MIC (mg/mL)			
		*k_cat_* (s^−1^)	*k_m_* (µM)	*k_cat_*/*k_m_* (µM^−1^ s^−1^)	Fold Increase ^a^	Biapenem	Meropenem	Imipenem	Ertapenem
OXA-23	-	0.015 ± 0.001	0.35	4.28 × 10^−2^	5.25	16	128	32	256
OXA-72 ^b^	-	0.013 ± 0.0005	0.22	5.80 × 10^−2^	7.11	32	128	64	512
OXA-66	OXA-51 (T^5^A, E^36^V, V^48^A, Q^107^K, P^194^Q, D^225^N)	0.03 ± 0.002	3.75	8.15 × 10^−3^	1.00	0.25	0.5	0.5	4
OXA-66 (K209M)	OXA-66 (K209M)	0.004 ± 0.0010	1.23	3.25 × 10^−3^	0.40	0.5	1	1	16
OXA-79	OXA-66 (W222G)	0.018 ± 0.0008	0.72	2.50 × 10^−2^	3.07	4	16	4	128
OXA-79 (I129L)	OXA-66 (W222G, I129L)	0.007 ± 0.0010	0.15	4.67 × 10^−2^	5.72	8	32	8	256
OXA-79 (K209M)	OXA-66 (W222G, K209M)	0.016 ± 0.0008	1.21	1.3 × 10^−2^	1.37	2	8	2	32
OXA-79 (W222M)	OXA-66 (W222M)	0.010 ± 0.0009	0.18	5.55 × 10^−2^	5.86	16	64	8	256
OXA-82	OXA-66 (L167V)	0.009 ± 0.0008	0.85	1.08 × 10^−2^	1.33	1	1	1	4
OXA-82 (I129L)	OXA-66 (L167V, I129L)	0.009 ± 0.0009	0.21	4.30 × 10^−2^	5.27	16	64	8	128
OXA-82 (I129L)	OXA-66 (L167V, I129L)	-	-	-	-	4	8	4	64
OXA-82 (K209M)	OXA-66 (L167V, K209M)	0.009 ± 0.0008	1.54	5.84 × 10^−3^	0.72	0.125	0.25	0.25	1
OXA-82 (W222G)	OXA-66 (L167V, W222G)	0.010 ± 0.0006	0.67	1.49 × 10^−2^	1.83	4	8	4	32
OXA-82 (W222M)	OXA-66 (L167V, W222M)	0.021 ± 0.0005	0.62	3.38 × 10^−2^	3.57	8	64	8	128
OXA-82 (I129L, W222G)	OXA-66 (L167V, I129L, W222G)	0.020 ± 0.0010	0.37	5.40 × 10^−2^	6.62	32	128	32	256
OXA-83	OXA-66 (I129L)	0.018 ± 0.0004	0.49	3.67 × 10^−2^	4.50	8	128	32	256
OXA-66 (I129A)	OXA-66 (I129A)	-	-	-	-	8	32	8	256
OXA-66 (I129D)	OXA-66 (I129D)	-	-	-	-	1	2	1	32
OXA-66 (I129F)	OXA-66 (I129F)	-	-	-	-	4	16	4	128
OXA-66 (I129M)	OXA-66 (I129M)	-	-	-	-	2	16	4	64
OXA-66 (I129V)	OXA-66 (I129V)	-	-	-	-	8	32	8	256
OXA-83 (W222G)	OXA-66 (I129L, W222G)	0.023 ± 0.0011	0.39	5.89 × 10^−2^	7.22	8	16	8	256
OXA-83 (W222M)	OXA-66 (I129L, W222M)	0.042 ± 0.0011	0.36	1.16 × 10^−1^	12.27	64	256	128	>512
OXA-99	OXA-51 (Q57R, K209M)	0.019 ± 0.0009	2.7	7.03 × 10^−3^	0.86	0.125	0.25	0.25	4
OXA-99 (I129L)	OXA-51 (Q57R, K209M, I129L)	0.021 ± 0.0007	1.63	1.29 × 10^−2^	1.58	2	4	2	32
Control ^c^	-	-	-	-	-	≤0.06	0.125	0.125	0.125
Control ^d^	-	-	-	-	-	≤0.06	≤0.06	≤0.06	≤0.06

^a^ The fold increases value of different variants with OXA-66 as reference. ^b^ OXA-72 is a variant of OXA-24/40. ^c^
*A. baumannii* strain ATCC17978. ^d^
*E. coli* strain ATCC25922.

## Data Availability

The data presented in this study are available in insert article.

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
