# Peer review of "Specific Amino Acid Substitutions in OXA-51-Type β-Lactamase Enhance Catalytic Activity to a Level Comparable to Carbapenemase OXA-23 and OXA-24/40"

_ijms, 2022, doi:10.3390/ijms23094496_

Round 1

Reviewer 1 Report

The authors took into account all the comments and improved the manuscript. The article can be published.

Reviewer 2 Report

The research is about a novel mechanism of resistance. It provides new data on the specific country and new epidemiological evidence compared with other published material.  The paper is well written, and the text is clear and easy to read. The conclusions are consistent with the evidence and arguments presented. It is original and well-done research

This manuscript is a resubmission of an earlier submission. The following is a list of the peer review reports and author responses from that submission.

Round 1

Reviewer 1 Report

This manuscript written by Chan BKW et al studied for enzymological characterization of OXA-51 valiants and point mutants of them. Furthermore, docking models of OXA-51 and meropenem were constructed, and interaction between the enzyme and substrate in the model were discussed with the enzyme parameters. 

The enzyme parameters were characterized well, but discussion for structure-function relationship were mess and nothing novel. Crystal structure of OXA-51 was determined by Smith et al (ref. 21). The reference paper stated that Trp222 in the native structure conflicted with superimposed carbapenem. In addition, the paper also demonstrated that Trp222Met in OXA-51 significantly decreased the Km. Taking together, Smith et al concluded that "The only way carbapenems can productively bind to the active site of OXA-51 is by displacing the side chain of Trp222 from the position it occupies in the current apo-OXA-51 structure." In this manuscript, however, a docking model between the native OXA-51 and meropenem was constructed without flipping of the Trp. As the result, the substrate in the model in Fig 2A was far from the active residue Ser80. Namely, the model does not reflect the structure mimicking the enzyme catalysis, while the model for Trp222Gly and meropenem is seems to be reasonable (it means they are not comparable). Thus, the discussion for structure-function relationship based on the model is non-sense. 

Author Response

Comments from Reviewer 1

This manuscript written by Chan BKW et al studied for enzymological characterization of OXA-51 valiants and point mutants of them. Furthermore, docking models of OXA-51 and meropenem were constructed, and interaction between the enzyme and substrate in the model were discussed with the enzyme parameters. 

The enzyme parameters were characterized well, but discussion for structure-function relationship were mess and nothing novel. Crystal structure of OXA-51 was determined by Smith et al (ref. 21). The reference paper stated that Trp222 in the native structure conflicted with superimposed carbapenem. In addition, the paper also demonstrated that Trp222Met in OXA-51 significantly decreased the Km. Taking together, Smith et al concluded that "The only way carbapenems can productively bind to the active site of OXA-51 is by displacing the side chain of Trp222 from the position it occupies in the current apo-OXA-51 structure." In this manuscript, however, a docking model between the native OXA-51 and meropenem was constructed without flipping of the Trp. As the result, the substrate in the model in Fig 2A was far from the active residue Ser80. Namely, the model does not reflect the structure mimicking the enzyme catalysis, while the model for Trp222Gly and meropenem is seems to be reasonable (it means they are not comparable). Thus, the discussion for structure-function relationship based on the model is non-sense. 

Response: Thank you for your suggestions. We have revised the orientation of W222 residue of OXA-51 and redo the docking again.

Reviewer 2 Report

The work of Chen and coworkers is based on the evaluation of the punctual contribution of amino acid residues in the catalysis of carbapenem-hydrolyzing class D  beta-lactamases (CHDLs). Variants of this enzyme are involved in the mediation of carbapenem resistance in Gram-negative bacteria. The goal of the study is to provide suggestions in order to understand the role of specific aminoacids behind the resistance.

The paper is well-written and understandable in all its sections. The results are clearly presented and linearly discussed. Moreover, the outcomes can stimulate further theoretical and experimental investigations and interest of a broad range of readers of the scientific community. I do not have major comments for this article and I’m favor to the publication on IJMS.

Three minor comments are reported as follows:

  • The title “Conclusion” for the conclusions section Is missing. I would include it in the article.
  • I’m not a fan of the syntax proposed by the authors, concerning the aminoacids name, see for instance W222, I129 I would prefer W222 and I129. Of course it is a matter of taste, but I find this kind of nomenclature and more understandable and I think that could improve the readability of the manuscript.
  • Please, provide a higher resolution version of the Figure 2, at page 12.

Author Response

Comments from Reviewer 2

The work of Chen and coworkers is based on the evaluation of the punctual contribution of amino acid residues in the catalysis of carbapenem-hydrolyzing class D  beta-lactamases (CHDLs). Variants of this enzyme are involved in the mediation of carbapenem resistance in Gram-negative bacteria. The goal of the study is to provide suggestions in order to understand the role of specific aminoacids behind the resistance.

The paper is well-written and understandable in all its sections. The results are clearly presented and linearly discussed. Moreover, the outcomes can stimulate further theoretical and experimental investigations and interest of a broad range of readers of the scientific community. I do not have major comments for this article and I’m favor to the publication on IJMS.

Three minor comments are reported as follows:

The title “Conclusion” for the conclusions section Is missing. I would include it in the article.

Response: Thank you for your comment. I have added the title in the text.

I’m not a fan of the syntax proposed by the authors, concerning the aminoacids name, see for instance W222, I129 I would prefer W222 and I129. Of course it is a matter of taste, but I find this kind of nomenclature and more understandable and I think that could improve the readability of the manuscript.

Response: Thank you for your suggestions. The syntax has been modified.

Please, provide a higher resolution version of the Figure 2, at page 12.

Response: Thank you for your comment. The resolution of Figure 2 has been increased.

Reviewer 3 Report

The work that is presented in the manuscript entitled ‘Specific amino acid substitutions in OXA-51-type β-lactamase enhance catalytic activity to a level comparable to carbapenemase OXA-23 and OXA-24/40’ by Chan et al. is interesting and clearly presented. The article style, however, should be reviewed in quite a lot points. Thus, I believe that the text needs some technical adjustments to be published. Therefore, I recommend that this manuscript can be published in IJMS after Major Revision.

My suggestions to the authors are listed below as “General comments” and “Specific comments”. Some comments are impossible to be made clearer since numbering of the lines is missing.   

General comments:

- In some parts the text is too lengthy and flow of the English language is lacking. I suggest to re-organize the text flow in smaller paragraphs to make the text more readable.

- Please use numbered sections headlines, and not only titles.

- Please double check reference list and provide references according to the MDPI style.

- Check the whole text for double – spacing, as it can be read in some parts.

- Please use space before citing a reference at the end of a sentence. Also, use space between a value and its units.

- Please use “s” for seconds and “h” for hours in terms of consistency.

- When referring to Tables and Figures in the main text, bold is not needed.

- Distinct Conclusion section is missing.

- Author contribution section is missing.

- Conflicts of interest section is missing.

Specific comments:

- Author affiliations: Please include the mails of all authors, according to MDPI style.

- Keywords: Enrich keywords list avoiding using the same words as in the title.

- Introduction section is too short. It needs enrichment in terms of background research and establishment of the novelty of the work.

- Materials and Methods, Antibiotics and media: Avoid using “while”. Please delete it.

- Materials and Methods, Structure modeling and Analysis: Please format “Schroedinger, Inc” as plain text.

- Table 2: Is it Km or km? Please be consistent. If different, please define. This also applies for Table 3, and corresponding discussion in text. Please, replace “≦” by “≤”,

- Table 2, footnote: what it is meant with “not done”. Add discussion.

- Table 3, footnote: Avoid using @, $, #,* symbols, better use letters of the Latin alphabet.

- Acknowledgment: this section should be renamed to “Funding”.

Author Response

Comments from Reviewer 3

My suggestions to the authors are listed below as “General comments” and “Specific comments”. Some comments are impossible to be made clearer since numbering of the lines is missing.    

General comments:

- In some parts the text is too lengthy and flow of the English language is lacking. I suggest to re-organize the text flow in smaller paragraphs to make the text more readable.

Response: Thank you for your comments. We have split the text into smaller paragraphs and subheadings are added for easy reading.

- Please use numbered sections headlines, and not only titles.

Response: Thank you for your comments. Numbers have been added to all section headlines.

- Please double check reference list and provide references according to the MDPI style.

Response: Thank you for your comment. The references have been changed to MDPI style.

- Check the whole text for double – spacing, as it can be read in some parts.

Response: Thank you for your comment. The double spacing were deleted.

- Please use space before citing a reference at the end of a sentence. Also, use space between a value and its units.

Response: Thank you for your comment. Suitable spacings have been added.

- Please use “s” for seconds and “h” for hours in terms of consistency.

Response: Thank you for your comment. The text has been changed.

- When referring to Tables and Figures in the main text, bold is not needed.

Response: Thank you for your comment. Bold has been removed.

- Distinct Conclusion section is missing.

- Author contribution section is missing.

- Conflicts of interest section is missing.

Response: Thank you for your comments. The conclusion is available in section 5 of the main text. The author’s contribution and declaration of conflict of interest has been added.

Specific comments:

- Author affiliations: Please include the mails of all authors, according to MDPI style.

Response: Thank you for your comments. The email address of all authors has been added.

- Keywords: Enrich keywords list avoiding using the same words as in the title.

Response: Thank you for your comments. The keyword list has been revised

- Introduction section is too short. It needs enrichment in terms of background research and establishment of the novelty of the work.

Response: Thank you for your comments. The extra background of this research has been added.

- Materials and Methods, Antibiotics and media: Avoid using “while”. Please delete it. 

Response: Thank you for your comments. The word “while” has been deleted.

- Materials and Methods, Structure modeling and Analysis: Please format “Schroedinger, Inc” as plain text.

Response: Thank you for your comments. The words has been changed to plain text.

- Table 2: Is it Km or km? Please be consistent. If different, please define. This also applies for Table 3, and corresponding discussion in text. Please, replace “≦” by “≤”,

Response: Thank you for your comments. The symbol has been changed.

- Table 2, footnote: what it is meant with “not done”. Add discussion.

Response: The footnote for sign “-“ has been changed to no data. Since it is the quality control strains for MIC test, it was impossible to conduct enzyme kinetics for them.

- Table 3, footnote: Avoid using @, $, #,* symbols, better use letters of the Latin alphabet.

Response:Thank you for your comments. The footnote has been modified.

- Acknowledgment: this section should be renamed to “Funding”.

Response: Thank you for your comments. The section has been revised to Funding.

Round 2

Reviewer 1 Report

Smith et al (ref. 21) stated, the side chain of Trp222 should be flipped to open the catalytic site, otherwise the substrate could not to be bound in the catalytic site. But the flipped  structure is not determined. If we can modify a structure on purpose, we can do whatever we want. Then, how to validate the structure? 

Author Response

Dear reviewer,

Thank you for your message.

A similar structure of flipped Trp222 could be found in the following reference: June, C. M., Muckenthaler, T. J., Schroder, E. C., Klamer, Z. L., Wawrzak, Z., Powers, R. A., Szarecka, A., & Leonard, D. A. (2016). The structure of a doripenem-bound OXA-51 class D β-lactamase variant with enhanced carbapenemase activity. Protein science : a publication of the Protein Society25(12), 2152–2163. https://doi.org/10.1002/pro.3040

In this article, the author analyzed the crystal structure of OXA-51 (I129L/K83D) with doripenem binding (PDB 5L2F). The structure showed a flipped on Trp222 residue. So we used this structure as the reference for our in silico prediction. We changed the OXA-51 (I129L/K83D) to wild-type OXA-51 and removed the molecule of doripenem, then docked the meropenem molecule into the OXA-51 protein. And the final result is shown in the main text of our updated manuscript.

Yours sincerely,

Bill

Reviewer 3 Report

Please clearly indicate the changes incorporated in the revised version of your manuscript.

Conclusion section is wrongly numbered as the third section, while it is the fifth section. Please review the numbering of the sections. 

Author Response

Dear Reviewer,

I apologize for the mistake I made that I have forgotten to upload the latest PDF of my updated manuscript. Therefore the numbering is wrong. I have updated the PDF file now.

For the changes, please refer to the updated word file. I have used the Track Changes function of MS word for easy tracking of the changes.

Thank you.

Yours sincerely,

Bill